# SIRT-1 is required for release of enveloped enteroviruses

Alagie Jassey, James Logue, Stuart Weston, Michael A Wagner, Ganna Galitska, Katelyn Miller, Matthew Frieman, William T Jackson*

Department of Microbiology and Immunology and Center for Pathogen Research, University of Maryland, Baltimore, Baltimore, United States

**Abstract** Enterovirus D68 (EV-D68) is a re-emerging enterovirus that causes acute respiratory illness in infants and has recently been linked to Acute Flaccid Myelitis. Here, we show that the histone deacetylase, SIRT-1, is essential for autophagy and EV-D68 infection. Knockdown of SIRT-1 inhibits autophagy and reduces EV-D68 extracellular titers. The proviral activity of SIRT-1 does not require its deacetylase activity or functional autophagy. SIRT-1's proviral activity is, we demonstrate, mediated through the repression of endoplasmic reticulum stress (ER stress). Inducing ER stress through thapsigargin treatment or SERCA2A knockdown in SIRT-1 knockdown cells had no additional effect on EV-D68 extracellular titers. Knockdown of SIRT-1 also decreases poliovirus and SARS-CoV-2 titers but not coxsackievirus B3. In non-lytic conditions, EV-D68 is primarily released in an enveloped form, and SIRT-1 is required for this process. Our data show that SIRT-1, through its translocation to the cytosol, is critical to promote the release of enveloped EV-D68 viral particles.

## eLife assessment

The presence or absence of a surrounding envelope, previously a clear distinguishing feature of different viruses, has been blurred by the recent recognition that many so-called 'nonenveloped' viruses are released from cells as quasi-enveloped virions cloaked in host cell membranes. This mechanism of viral egress allows for non-lytic infection, and has potentially **important** implications for pathogenesis. In this manuscript, Jassey and colleagues provide **solid** evidence that the protein deacetylase SIRT-1 is required for the non-lytic release of enteroviruses in extracellular vesicles.

## Introduction

Enterovirus D68 (EV-D68) is a re-emerging enterovirus and a cause of acute respiratory illness in infants. EV-D68 infection has recently been associated with Acute Flaccid Myelitis, a severe polio-like neurological disease that causes limb weakness and loss of muscle tone in infants (*Eshaghi et al., 2017*; *Lang et al., 2014*). There is currently no FDA-approved drug or vaccine against EV-D68. It is important to understand how EV-D68 hijacks host processes to facilitate its life cycle within the host.

EV-D68 is a positive-sense single-stranded RNA enterovirus belonging to the *Piconarviridae* family. The viral genome encodes a single polyprotein that is immediately processed upon translation by viral proteases into multiple intermediate and mature structural and seven non-structural proteins (*Esposito et al., 2015*). Enterovirus infection induces extensive rearrangement of cytosolic membranes, beginning with the formation of complex single convoluted membranes, also known as replication vesicles, on the cytosolic face of which viral RNA replication occurs. For most enteroviruses, these intricate structures eventually morph into double-membrane autophagosome-like structures, which play roles in virus replication, maturation, and non-lytic release (*Jackson, 2014*; *Jackson et al., 2005*; *Shi and Luo, 2012*).

**\*For correspondence:**
wjackson@som.umaryland.edu

**Competing interest:** The authors declare that no competing interests exist.

Autophagy is a highly regulated catabolic process that maintains cell survival by targeting superfluous cytoplasmic contents and infectious microorganisms, including pathogenic bacteria and viruses, for degradation. Several RNA viruses, including EV-D68, are known to subvert autophagy for their own benefit, and our interest in this pathway led us to study sirtuin 1 (SIRT-1) (*Ahmad et al., 2018*). SIRT-1 is important for autophagy initiation, but whether SIRT-1 is essential for the later stages of autophagy (autophagic flux) or translocates to the cytosol during starvation or other stress stimuli, such as viral infection, is unknown (*Bai and Zhang, 2016*; *Huang et al., 2015*; *Lee et al., 2008*). A growing body of evidence implicates SIRT-1 as an essential regulator of autophagy downstream of the nucleation complex (*Green and Levine, 2014*; *Huang et al., 2015*; *Lee et al., 2008*).

SIRT-1 belongs to the $NAD^+$-dependent family of histone deacetylase enzymes, which are known to control several physiological processes. The sirtuin family of enzymes contains seven members (SIRT-1–7), which display varying subcellular localization, with SIRT-1 being the most studied owing to its role in lifespan expansion (*Chalkiadaki and Guarente, 2015*; *Jing and Lin, 2015*). SIRT-1 has been shown to regulate cellular responses to various stresses, including the cell cycle, apoptosis, inflammation, endoplasmic reticulum stress (ER stress), and more (*Chalkiadaki and Guarente, 2015*). SIRT-1 has also been shown to promote the Middle East respiratory syndrome coronavirus infection, but whether SIRT-1 regulates picornavirus infection is unknown (*Weston et al., 2019*).

A less well-known or studied role for SIRT-1 is in regulating ER stress (*Li et al., 2014*; *Singh et al., 2020*; *Tang et al., 2018*). ER stress, which can be provoked by various stress stimuli, including viral infection, happens when proteins cannot properly fold, leading to the accumulation of unfolded/misfolded protein. ER stress engages the unfolded protein response (UPR), which attenuates transcription and translation, decreases protein synthesis, and enhances the expression of molecular chaperones to increase the cell's protein folding capacity (*Oslowski and Urano, 2011*). The inositol-requiring enzyme (IRE) 1-X-box-binding protein (XBP) 1 pathway constitutes one of the three major pathways induced by UPR (*Hetz, 2012*). Upon ER stress, IRE-1 cleaves XBP-1 mRNA into spliced forms which, in turn, activates UPR target genes (*Yoshida et al., 2001*). Studies have shown an intricate interplay between SIRT-1 and ER stress. While SIRT-1 regulates ER stress by binding and inhibiting the transcriptional activity of XBP1, ER stress also regulates SIRT-1 levels (*Wang et al., 2011*). For instance, ER stress induction through thapsigargin (TG) treatment induces SIRT-1 protein levels in vivo and in vitro to promote hepatocellular injury (*Koga et al., 2015*).

Here, we report that EV-D68 infection induces SIRT-1 translocation to the cytosol. We also show that SIRT knockdown impedes the extracellular vesicle-mediated release of infectious EV-D68 particles. We further demonstrate that the pro-EV-D68 activity of SIRT-1 is mediated through repressing ER stress and does not require SIRT-1's deacetylase function, nor is it dependent on functional autophagy. Moreover, SIRT-1 knockdown reduces extracellular titers of poliovirus (PV) and SARS-CoV-2 but not coxsackievirus B3 (CVB3). Our results indicate that certain enteroviruses induce SIRT-1 translocation to the cytosol and require the cellular protein for efficient release.

## Results

### EV-D68 infection changes SIRT-1's subcellular localization

SIRT-1 is proviral in some cases and has previously been reported to regulate autophagy initiation, which is important for EV-D68 infection (*Huang et al., 2015*; *Lee et al., 2008*; *Weston et al., 2019*). But whether SIRT-1 is essential for EV-D68 reproduction is unknown. We first interrogated the effects of SIRT-1 knockdown on EV-D68 titers. SIRT-1 knockdown reduced both the intracellular (*Figure 1A*) and extracellular (*Figure 1B*) EV-D68 titers, with the knockdown being more effective in decreasing the extracellular titers than the cell-associated viral titers. Pharmacological inhibition of SIRT-1, through EX527 pretreatment, yielded similar results (*Figure 1—figure supplement 1A, B*). Next, we assessed the effect of SIRT-1 overexpression on EV-D68 titers. Both wild-type (SIRT-1 WT) and the deacetylase inactive mutant (SIRT-1 H363Y) increased EV-D68 extracellular titers (*Figure 1C*). Furthermore, overexpression of both wild-type and mutant SIRT-1 constructs partially rescued EV-D68 extracellular titers in SIRT-1 knockdown cells (*Figure 1—figure supplement 1C*), indicating that the pro-EV-D68 activity of SIRT-1 does not require its deacetylase function.

Given SIRT-1's proviral role in EV-D68 infection, we asked whether EV-D68 infection alters SIRT-1 protein levels or subcellular localization. We infected H1HeLa cells for various time points before

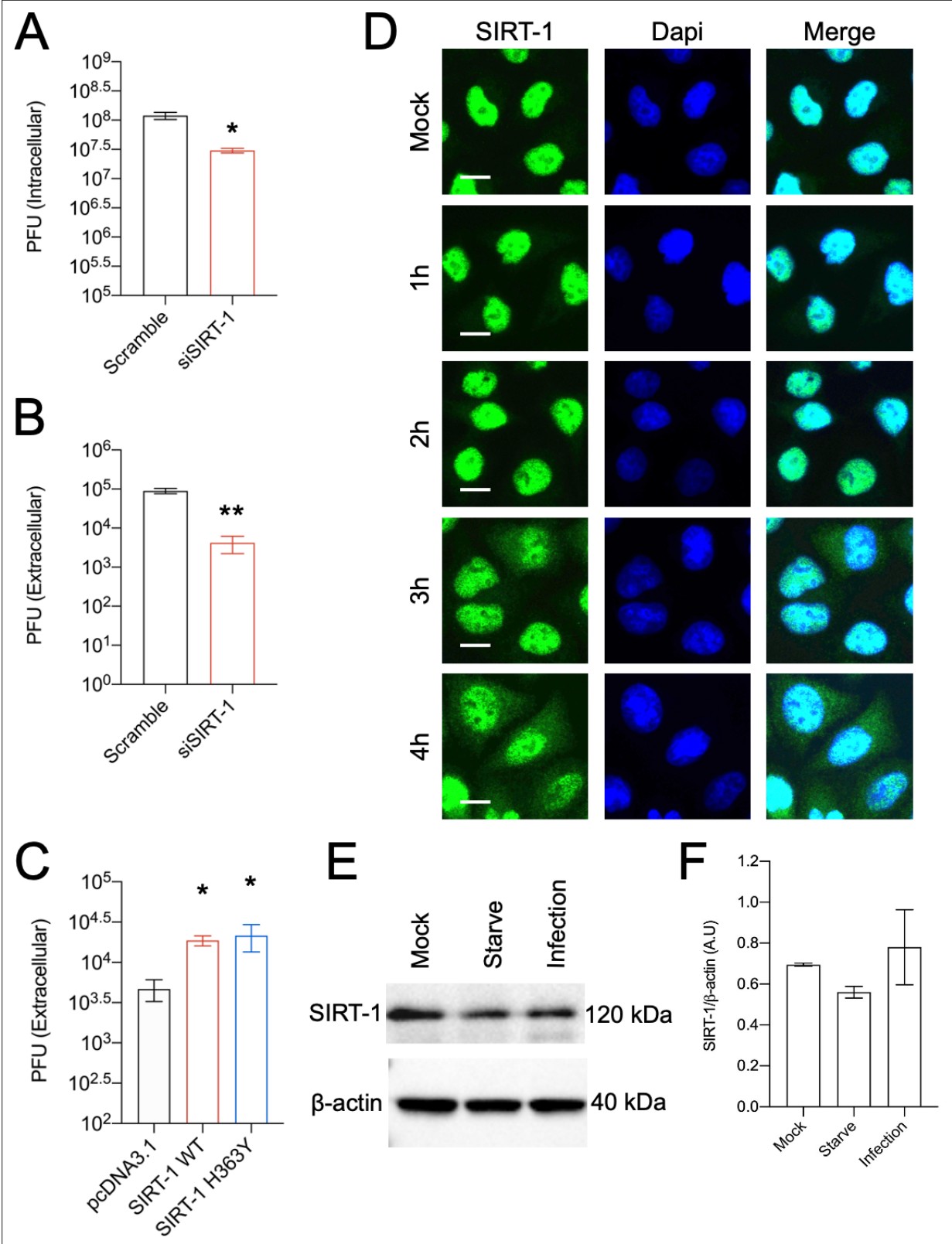

**Figure 1.** Enterovirus D68 (EV-D68) infection changes SIRT-1's subcellular localization. H1HeLa cells were transfected with either scramble control or SIRT-1 siRNAs for 48 hr. The cells were then infected with EV-D68 (multiplicity of infection [MOI = 0.1]) for 5 hr. The intracellular (**A**) and extracellular (**B**) particles were collected for plaque assay. (**C**) Cells were transfected with the indicated plasmids for 24 hr before being infected with EV-D68 (MOI = 0.1) for 5 hr. The extracellular particles were collected and analyzed by a plaque assay. (**D**) H1HeLa cells were infected for the indicated time points for

*Figure 1 continued on next page*

Figure 1 continued

immunofluorescence analysis (IFA) against SIRT-1. (**E**) H1HeLa cells were infected (MOI = 30) for 4 hr. Lysates were collected for western blot against SIRT-1. (**F**) Densitometry quantitation of **E**. Error bars denote the mean ± standard error of the mean (SEM) of three independent repeats. Unpaired Student's $t$-test was used for the statistical analyses (**p < 0.01; *p ≤ 0.05; ns = not significant). Scale bar = 6.2 µm.

The online version of this article includes the following figure supplement(s) for figure 1:

**Figure supplement 1.** SIRT-1 in LC3 puncta formation and virus release.

**Figure supplement 2.** Leptomycin B (LMB) inhibits Transcription Factor EB (TFEB) nuclear export but not SIRT-1 translocation during enterovirus D68 (EV-D68) infection.

**Figure supplement 3.** Enterovirus D68 (EV-D68) infection does not alter RB1CC1 subcellular localization.

subjecting the cells to immunofluorescence analysis (IFA) against SIRT-1. As indicated in *Figure 1D*, SIRT-1 is localized to the nucleus in the mock-infected cells. EV-D68 infection, on the other hand, induced translocation of a fraction of SIRT-1 to the cytosol (*Figure 1D*) beginning at 3 hpi, which coincides with peak viral RNA replication. Interestingly, EV71 has also been reported to induce SIRT-1 relocalization to the cytosol during infection (*Han et al., 2016*). To understand whether SIRT-1 translocation during EV-D68 infection is dependent on exportin-1, the major mammalian nuclear export protein, we pretreated H1HeLa cells with and without leptomycin-B, a specific exportin-1 inhibitor, followed by EV-D68 infection (*Fornerod et al., 1997*; *Fukuda et al., 1997*). Leptomycin-B treatment inhibited transcription factor-EB, the master transcriptional regulator of autophagy translocation upon refeeding, as expected (*Figure 1—figure supplement 2A*), but it did not obstruct SIRT-1 translocation induced by EV-D68 infection (*Figure 1—figure supplement 2B*). In some cases, cleavage of nuclear pore proteins by enteroviruses has been shown to cause leakage of proteins from the nucleus during infection (*Gustin and Sarnow, 2002*; *Hanson et al., 2019*; *Watters et al., 2017*). However, we found that the autophagy kinase regulator RB1CC1 does not leave the nucleus during EV-D68 infection, indicating that SIRT-1 is being targeted for translocation (*Figure 1—figure supplement 3*). We then examined the effect of EV-D68 infection on SIRT-1 protein levels. Results in *Figure 1E* and its associated densitometry analysis (*Figure 1F*) revealed that in contrast to starvation, which marginally decreased SIRT-1 protein levels, EV-D68 infection does not significantly impact SIRT-1 protein levels as has been reported for other enteroviruses (*Han et al., 2016*; *Kanda et al., 2015*; *Xander et al., 2019*).

## SIRT-1 promotes autophagy but decreases EV-D68 extracellular titers in autophagy-deficient ATG-7 KO cells

While SIRT-1 is known to regulate autophagy initiation, whether the cellular histone deacetylase is important for basal and stress-induced autophagy is unclear. SIRT-1 and scramble knockdown H1HeLa cells were starved or treated with carbonyl cyanide *m*-chlorophenylhydrazone (CCCP) to induce or inhibit autophagy, respectively. As shown in *Figure 2A*, starvation reduces p62 protein levels, and CCCP treatment increased LC3 lipidation in the scramble control group as expected. In contrast, acute amino acid starvation did not significantly alter p62 protein expression, and CCCP treatment failed to trigger LC3II accumulation in SIRT-1 knockdown cells, confirming the published work that SIRT-1 is essential for basal and starvation-induced autophagy. To further test SIRT-1's importance for basal autophagy, we knocked down SIRT-1 and performed IFA against endogenous LC3.

In contrast to the scramble control, which mainly displayed diffused nuclear-localized LC3, SIRT-1 knockdown provoked endogenous LC3 puncta accumulation near the perinuclear region (*Figure 2B*). Similar results were observed in GFP-LC3-overexpressing SIRT-1 knockdown cells, wherein knockdown of SIRT triggered GFP-LC3 puncta accumulation (*Figure 1—figure supplement 1E*). Together, these results show that SIRT-1 is essential for stress-induced and basal autophagy.

Given SIRT-1's regulation of autophagy, we asked whether SIRT-1's proviral activity depends on functional autophagy. SIRT-1, which is typically localized to the nucleus in cancer cells, forms a molecular complex with three proteins essential for autophagy initiation: ATG-5, ATG-7, and LC3. SIRT-1 deacetylates these autophagy-related proteins, thereby regulating autophagy induction (*Bai and Zhang, 2016*; *Lee et al., 2008*). We first examined whether SIRT-1 colocalizes with ATG-7 during EV-D68 infection. As shown in *Figure 2E*, EV-D68 induces ATG-7 puncta formation, which partially colocalizes with SIRT-1. We next examined the effect of ATG-7 knockout (ATG-7 KO), which cannot

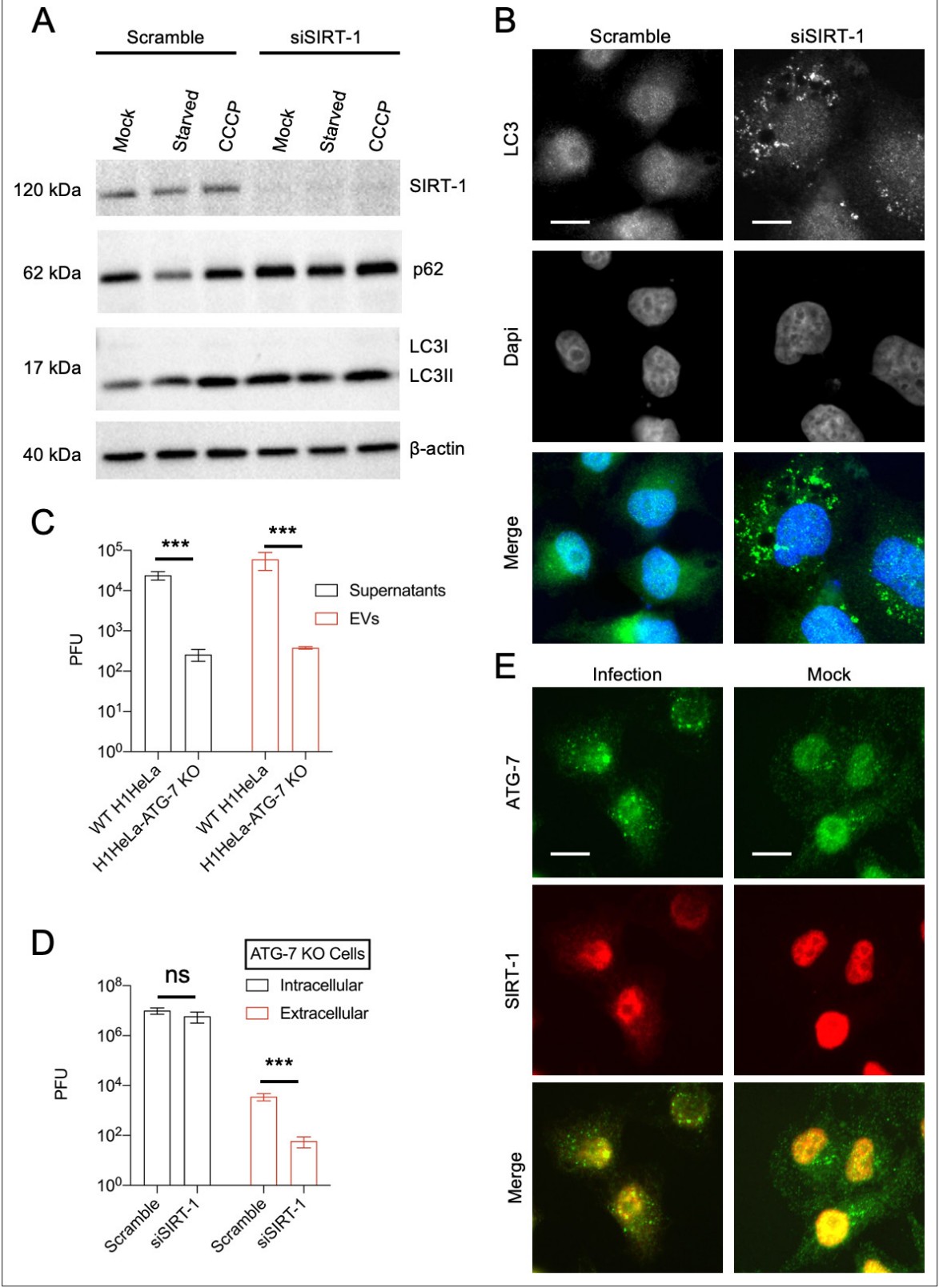

**Figure 2.** SIRT-1 promotes autophagy but decreases enterovirus D68 (EV-D68) extracellular titers in autophagy-deficient ATG-7 KO cells. (**A**) H1HeLa cells were transfected with scramble or SIRT-1 siRNA for 48 hr. The cells were subsequently starved or treated with CCCP (10 µM) for 4 hr. Lysates were harvested and analyzed by western blot. (**B**) Cells were transfected with the indicated siRNAs for 48 hr before being fixed and subjected to immunofluorescence analysis (IFA) against endogenous LC3. (**C**) H1HeLa and ATG-7 KO cells were infected with 0.1 MOI of EV-D68 for 5 hr. The

*Figure 2 continued on next page*

*Figure 2 continued*

extracellular vesicles were isolated as described in the materials and methods and viral titers were determined by a plaque assay. (**D**) ATG-7 KO cells were transfected with scramble or SIRT-1 siRNAs for 48 hr and infected as in C, followed by plaque assay-based viral titer determination. (**E**) H1Hela cells were infected with EV-D68 (MOI = 30) for 4 hr. The cells were fixed and immuno-stained with antibodies against SIRT-1 and ATG-7. Error bars denote the mean ± standard error of the mean (SEM) of three independent repeats. Unpaired Student's *t*-test was used for the statistical analyses (***p < 0.001; ns = not significant). Scale bar = 6 μm.

The online version of this article includes the following figure supplement(s) for figure 2:

**Figure supplement 1.** SIRT-1 does not colocalize with p62 during starvation.

form autophagosomes, on EV-D68 release. ATG-7 KO severely impeded both extracellular and vesicle-mediated release of EV-D68 (***Figure 2C***), consistent with the importance of autophagy for picornavirus release (***Jackson, 2014***). We then knocked down SIRT-1 in ATG-7 knockout (ATG-7 KO) cells, then infected these cells with EV-D68. Knockdown of SIRT-1 decreased EV-D68 extracellular titers in ATG-7 KO cells without significantly altering the intracellular titers (***Figure 2D***), suggesting that the proviral function of SIRT-1 does not require functional autophagy. Since SIRT-1 regulates autophagy through its deacetylase activity, these data are consistent with our findings in ***Figure 1***.

## SIRT-1's proviral activity is mediated by repressing ER stress

SIRT-1 has previously been reported to negatively regulate ER stress by deacetylating X-box-binding protein-1 (XBP1) (***Wang et al., 2011***). To determine whether SIRT-1's proviral activity is mediated via ER stress, we treated H1HeLa cells with or without TG, a well-known ER stress inducer, for 5 hr after viral adsorption. TG provokes ER stress by inhibiting the Sarco/endoplasmic reticulum $Ca^{2+}$-ATPase pump (SERCA), which mediates calcium transfer from the cytosol to the lumen of the ER, and is important for maintaining ER homeostasis. While TG treatment only marginally reduced EV-D68 intracellular titers, it markedly decreased the extracellular titers (***Figure 3A***), similar to what we observed in SIRT-1 knockdown cells (***Figure 1A, B***). This finding suggests that TG and SIRT-1 may share common cellular targets. To test this hypothesis, we treated scramble and SIRT-1 knockdown cells with and without TG after viral adsorption. TG treatment significantly reduced viral extracellular titers in the scramble control group. In contrast, TG did not significantly alter viral extracellular titers when SIRT-1 is depleted (***Figure 3B***). We also observed similar results in our genetic epistasis analysis, in which the knockdown of SERCA2A did not further reduce viral titers in SIRT-1-depleted cells (***Figure 3C, D***). We then examined the effect of SIRT-1 depletion and EV-D68 infection on SERCA2A protein levels by western blot. While EV-D68 infection did not alter SERCA2A levels, the knockdown of SIRT-1 decreases SERCA2A protein levels (***Figure 3E***), and overexpression of wild-type SIRT-1, not the deacetylase defective mutant, rescued SERCA2A protein levels, suggesting that SIRT-1's deacetylase activity is essential for SERCA2A stability (***Figure 1—figure supplement 1D***).

Since the induction of ER stress, through either SIRT-1 depletion or TG treatment, attenuates EV-D68 egress, we asked whether EV-D68 infection induces ER stress. For this purpose, H1HeLa cells were mock-infected, infected with EV-D68, or treated with TG for western blot against XBP1, which is upregulated upon ER stress. As expected, TG treatment, which was used as a positive control, induced XBP-1 protein levels in H1HeLa cells (***Figure 3F***). In contrast, EV-D68 infection reduced XBP1 levels compared to the mock-infection control (***Figure 3F***), indicating that the virus does not trigger ER stress in H1HeLa cells.

## Induction of ER stress impairs EV-D68 release in hSABCi-NS1.1 primary cells

To test whether TG could impair EV-D68 non-lytic release in a more physiologically relevant cell line, we infected hSABCi-NS1.1 immortalized small airway cells with and without TG. As shown in ***Figure 4A***, TG treatment significantly reduced EV-D68 extracellular titers without appreciably impacting the intracellular titers, similar to what we observed in H1HeLa cells (***Figure 3A***). Our western blot data in ***Figure 4B*** showed that while TG treatment induced BiP protein levels, indicating the induction of ER stress in hSABCi-NS1.1 cells, EV-D68 infection reduced BiP levels compared to the mock control and impaired TG-induced BiP protein expression, suggesting that EV-D68 reduces ER stress. Next, we examined the impact of EX527 pretreatment on EV-D68 titers in the primary cells. Similar to our

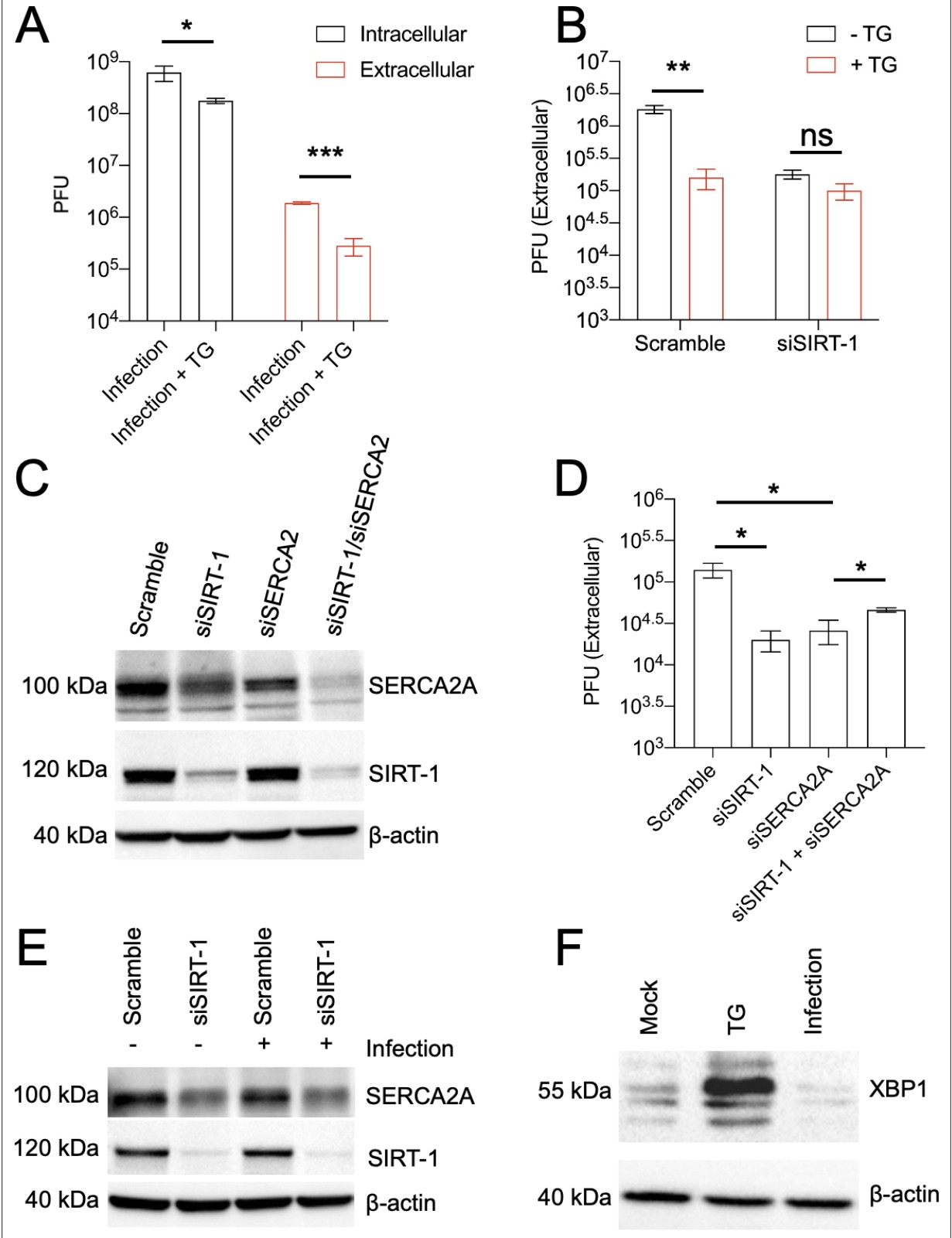

**Figure 3.** The proviral activity of SIRT-1 is mediated through endoplasmic reticulum stress (ER stress). (**A**) H1HeLa cells were infected with enterovirus D68 (EV-D68) (MOI = 0.1) for 30 min. The cells were washed and replenished with complete media with or without 2 μM thapsigargin (TG) for 5 hr. Viral titers were determined by plaque assay. (**B**) Cells were transfected with either scramble control siRNA or SIRT-1 siRNA for 48 hr. The cells were infected and treated as in A, and viral extracellular titers were similarly measured by plaque assay. Cells were transfected with the indicated siRNAs for 48 hr

*Figure 3 continued on next page*

*Figure 3 continued*

for western blot (**C**) or viral titer determination (**D**) following EV-D68 infection (MOI = 0.1) for 5 hr. (**E**) H1HeLa cells were transfected for 48 hr with the indicated siRNAs, and either mock-infected or infected (with EV-D68 MOI = 30) for 4 hr. Lysates were collected, and a western blot was performed against the indicated proteins. (**F**) H1HeLa cells were mock-infected, treated with TG for 4 hr, or infected with EV-D68 (MOI = 30) for 4 hr for western blot against XBP1. Error bars indicate mean ± standard error of the mean (SEM) of at least three independent experiments. Unpaired Student's *t*-test was used for the statistical analysis (***p < 0.001; **p < 0.01; *p ≤ 0.05; ns = not significant).

observation in H1HeLa cells, EX527 reduced EV-D68 extracellular titers and only marginally decreased EV-D68 intracellular titers (*Figure 4C*). Finally, we inquired whether EV-D68 infection causes SIRT-1 translocation in these primary cells. As indicated in *Figure 4D*, SIRT-1 localizes to the nucleus in uninfected cells. In contrast, EV-D68 infection, as shown by VP3 staining, induces SIRT-1 translocation to the cytosol. These findings demonstrate that SIRT-1's proviral effect in mediating EV-D68 non-lytic release is not cell-type specific.

## SIRT-1 reduces PV, not CVB3, titers

We next sought to understand whether SIRT-1 modulates the infection of other medically important enteroviruses, including PV and CVB3. While protective vaccines are available for PV, few vaccine-derived PV infection cases exist. On the other hand, there is no vaccine or treatment against CVB3, a significant cause of myocarditis and neurological disorders in infants. Therefore, identifying host factors that influence the infection of these viruses could help control their infection. We first examined whether PV and CVB3 alter the subcellular localization of SIRT-1. To our surprise, while PV, similar to EV-D68, induces SIRT-1 translocation to the cytosol, CVB3 did not significantly alter SIRT-1's subcellular localization (*Figure 5A*). We then examined the impact of SIRT-1 knockdown on PV and CVB3 infection. SIRT-1 knockdown marginally reduced PV intracellular titers but significantly decreased its extracellular titers (*Figure 5B, C*). In contrast, the knockdown of SIRT-1 did not alter CVB3 intracellular titers but slightly increased CVB3 extracellular titers (*Figure 5D, E*). To further analyze SIRT-1's role in PV and CVB3 infection, we pretreated H1HeLa cells with EX527 followed by viral infection. Consistent with siRNA-mediated depletion of SIRT-1, EX527 decreased PV titers, but not CVB3 titers (*Figure 5—figure supplement 1*). These results indicate that some, but not all, picornaviruses require SIRT-1 for their egress from infected cells.

## Knockdown of SIRT-1 reduces SARS-CoV-2 titers

SIRT-1 was previously reported as essential for Middle Eastern respiratory syndrome (MERS) coronavirus infection (*Weston et al., 2019*). Given the current ongoing SARS-CoV-2 pandemic's significance and to identify host factors/processes important for SARS-CoV-2 infection, we asked whether SIRT-1 is necessary for SARS-CoV-2 infection. A549-ACE-2 cells were transfected with either scramble or SIRT-1 siRNA for 48 hr (*Figure 6B*), followed by SARS-CoV-2 infection for 24 hr. As shown in *Figure 6C*, the knockdown of SIRT-1 impeded SARS-CoV-2 release. We then examined the impact of SARS-CoV-2 infection on SIRT-1's subcellular localization. As indicated in *Figure 6A*, SARS-CoV-2 infection triggers SIRT-1 translocation to the cytosol. These results show that, as with MERS-CoV, SIRT-1 is essential for SARS-CoV-2 infection.

## SIRT-1 knockdown reduces the extracellular vesicle-mediated release of enveloped EV-D68 viral particles

Since SIRT-1 knockdown reduces the extracellular EV-D68 titer an order of magnitude more than the intracellular titers, we posit that SIRT-1 proviral activity promotes non-lytic viral release. Multiple members of the *Piconarviridae* family have been previously reported to be released in an enveloped form, usually with multiple virions per vesicle (*Chen et al., 2015*; *Feng et al., 2013*; *Robinson et al., 2014*; *Sin et al., 2017*). Hence, we asked whether EV-D68 is similarly released in extracellular vesicles (EVs) and whether SIRT-1 is essential for this process. We knocked down SIRT-1 and isolated EVs for viral titer measurement. As shown in *Figure 7A*, EV-D68 appears to be released chiefly in EVs compared to the post-spin supernatant (PSS), and SIRT-1 knockdown severely impeded the extracellular vesicle-mediated release of EV-D68. We knocked down SIRT-1 to understand its impact on viral release. We also examined the expression of CD63, the most widely used marker for exosomes/multivesicular bodies, by western blot. As depicted in *Figure 7B*, SIRT-1 knockdown induced CD63 protein levels

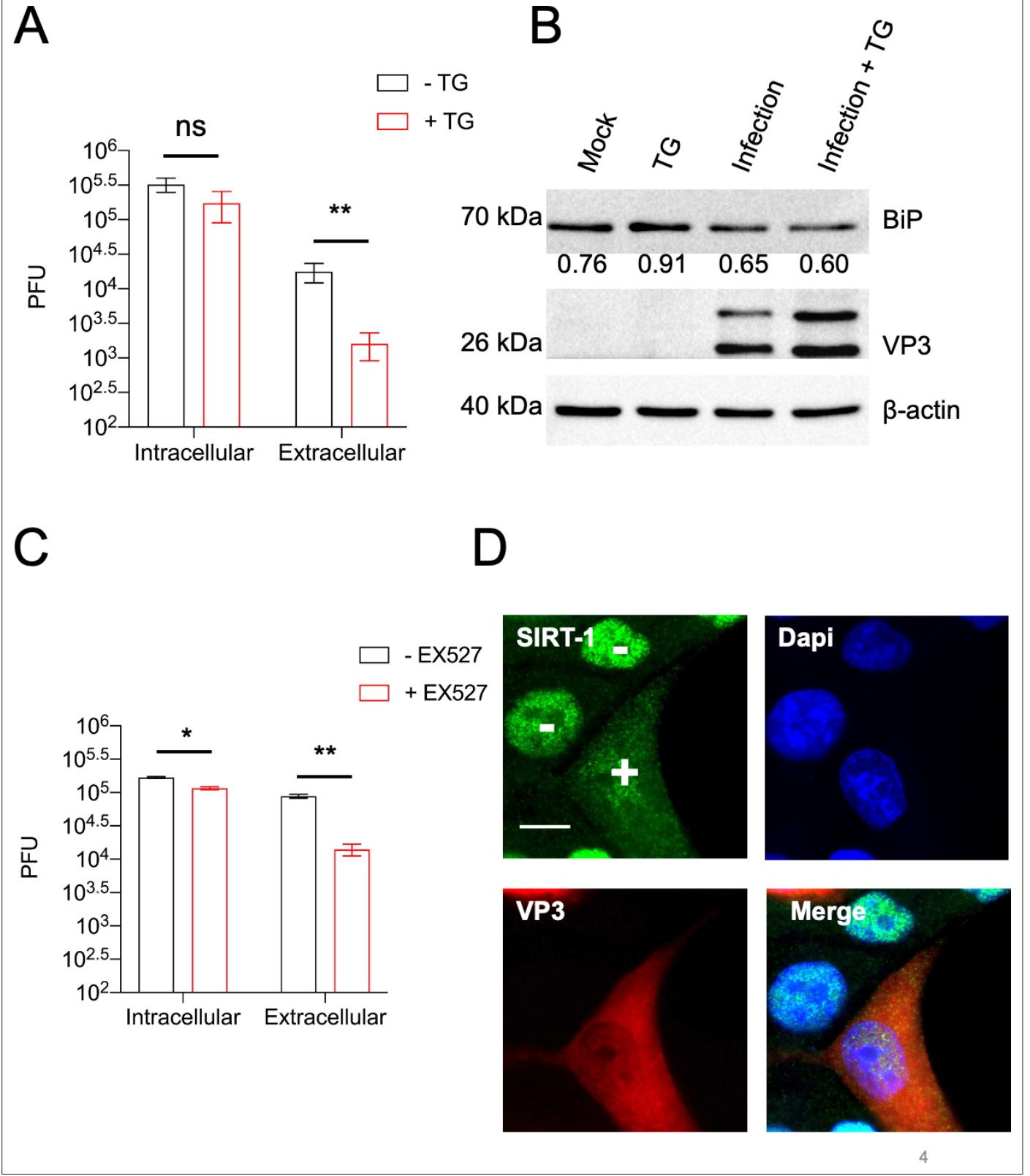

**Figure 4.** Induction of endoplasmic reticulum stress (ER stress) impairs enterovirus D68 (EV-D68) release in hSABCi-NS1.1 cells. (**A**) hSABCi-NS1.1 cells were infected with 0.1 MOI of EV-D68 for 30 min. The cells were washed and replenished with basal media or media containing 2 µM thapsigargin (TG) for 6 hr. Viral titers were determined by plaque assay. (**B**) hSABCi-NS1.1 cells were mock-infected, infected with EV-D68, and treated with or without TG for 6 hr, or treated with TG alone for 6 hr. The lysates were harvested and a western blot was performed against the indicated proteins. *n* = 2

*Figure 4 continued on next page*

*Figure 4 continued*

independent experiments. (**C**) hSABCi-NS1.1 were pretreated with 50 μM EX527 for 48 hr. The cells were then infected with EV-D68 (MOI = 0.1) for 6 hr for a plaque assay. (**D**) hSABCi-NS1.1 cells were mock-infected or infected with EV-D68 (MOI = 30) for 6 hr. The cells were fixed for immunofluorescence analysis (IFA) against endogenous SIRT-1. Error bars represent mean ± standard error of the mean (SEM) (**p < 0.01; *p ≤ 0.05; ns = not significant). Scale bar = 7.5 μm.

compared to the scramble control. Given the increase in CD63 protein levels in SIRT-1 knockdown cells, we hypothesize that the knockdown of SIRT-1 may prevent the release of CD63-positive EVs. To test this, we infected SIRT-1 knockdown cells for 4 hr and isolated the EVs for western blotting against CD63. As shown in *Figure 7C*, SIRT-1 knockdown increased CD63 levels in the whole-cell lysate. Our IFA analysis in *Figure 7D* also showed the aggregation of sizable CD63-positive puncta in SIRT-1 knockdown cells, as previously observed (*Latifkar et al., 2019*). Interestingly, for the extracellular vesicle fraction, we observed that SIRT-1 knockdown decreased the release of CD63-positive EVs during EV-D68 infection compared to the scramble control (*Figure 7C*). Consistent with the decrease in CD63-positive signal in SIRT-1 knockdown EVs, we detected VP3 only in the scramble control EVs, not EVs purified from SIRT-1 knockdown cells (*Figure 7C*). Together, these results indicate that SIRT-1 is essential for EV-D68 release in EVs.

## Discussion

Our work shows that SIRT-1, largely thought of as a transcriptional regulator, is essential for the release of the enveloped form of EV-D68. Here, we confirm existing data that SIRT-1 is essential for basal and starvation-mediated autophagy, and traffics to the cytosol during autophagic induction and infeciton. In addition, we show that most EV-D68 is released non-lytically in an enveloped form, and that knock-down or pharmacological inhibition of SIRT-1 severely decreases the release of infectious enveloped EV-D68 particles. Knockdown of SIRT-1 also attenuates the release of PV and SARS-CoV-2 but margin-ally increases CVB3 egress (*Figures 5C, E and 6C*). Our results suggest that many viruses induce SIRT-1 translocation to the cytosol to promote their vesicular release. Interestingly, our genetic and pharmacological data lead us to conclude that SIRT-1 is not acting to promote virus release through autophagy, nor through histone deacetylation, but through a mechanism related to ER stress.

We observed that EV-D68 infection induces relocalization of SIRT-1 from the nucleus to the cytosol starting at 3 hpi (*Figure 1D*). Given that RNA replication peaks at 3 hpi, we initially thought SIRT-1 might be essential for viral RNA replication. However, during EV-D68 infection, SIRT-1 did not colo-calize to dsRNA, a marker of active viral RNA replication (*Figure 7—figure supplement 1A*). In contrast, GBF-1, a cellular factor that is required for enterovirus replication, colocalizes with dsRNA during EV-D68 infection (*Figure 7—figure supplement 1B*). The kinetic studies in our supplemental data revealed that starvation induces rapid SIRT-1 translocation to the cytosol as early as 1 hr post-treatment (*Figure 2—figure supplement 1A*). This finding is consistent with the notion that SIRT-1 drives autophagosome formation during amino acid starvation. Although starvation slightly decreased SIRT-1 protein levels by western blot, pharmacological inhibition of autophagic flux failed to restore SIRT-1 levels. Moreover, SIRT-1 did not colocalize with p62 in starved cells (*Figure 2—figure supple-ment 1B*), suggesting that the cellular protein is not a substrate for autophagy.

We found that SIRT-1 knockdown did not significantly impact EV-D68 RNA replication (*Figure 7—figure supplement 1C*), virus binding (*Figure 7—figure supplement 1D*), or virus entry (*Figure 7—figure supplement 1E*). Instead, our findings suggest that the effect of SIRT-1 on virus production can largely be explained by a role in re-configuring the exocytosis pathway to promote the release of virus-loaded EVs. Knockdown of SIRT-1 led to the accumulation of large CD63-positive puncta in cells (*Figure 7D*), suggesting that the turnover, or the release, of these vesicles is downregulated in the absence of SIRT-1. Infection results in an increase in CD63 in the extracellular vesicles. Interest-ingly, our western blot data show a marked decrease in CD63 and a complete lack of VP3 detection in EVs from infected cells with reduced SIRT-1 expression (*Figure 7C*). This finding, coupled with the increased CD63 in the whole-cell lysates (*Figure 7B, C*), indicates that the reduction of SIRT-1 attenu-ates the release of virus-loaded CD63-positive EVs.

While enteroviruses are thought to be mainly released by cell lysis, a growing body of evidence indicates that these viruses can be released from intact cells without cell lysis, a phenomenon

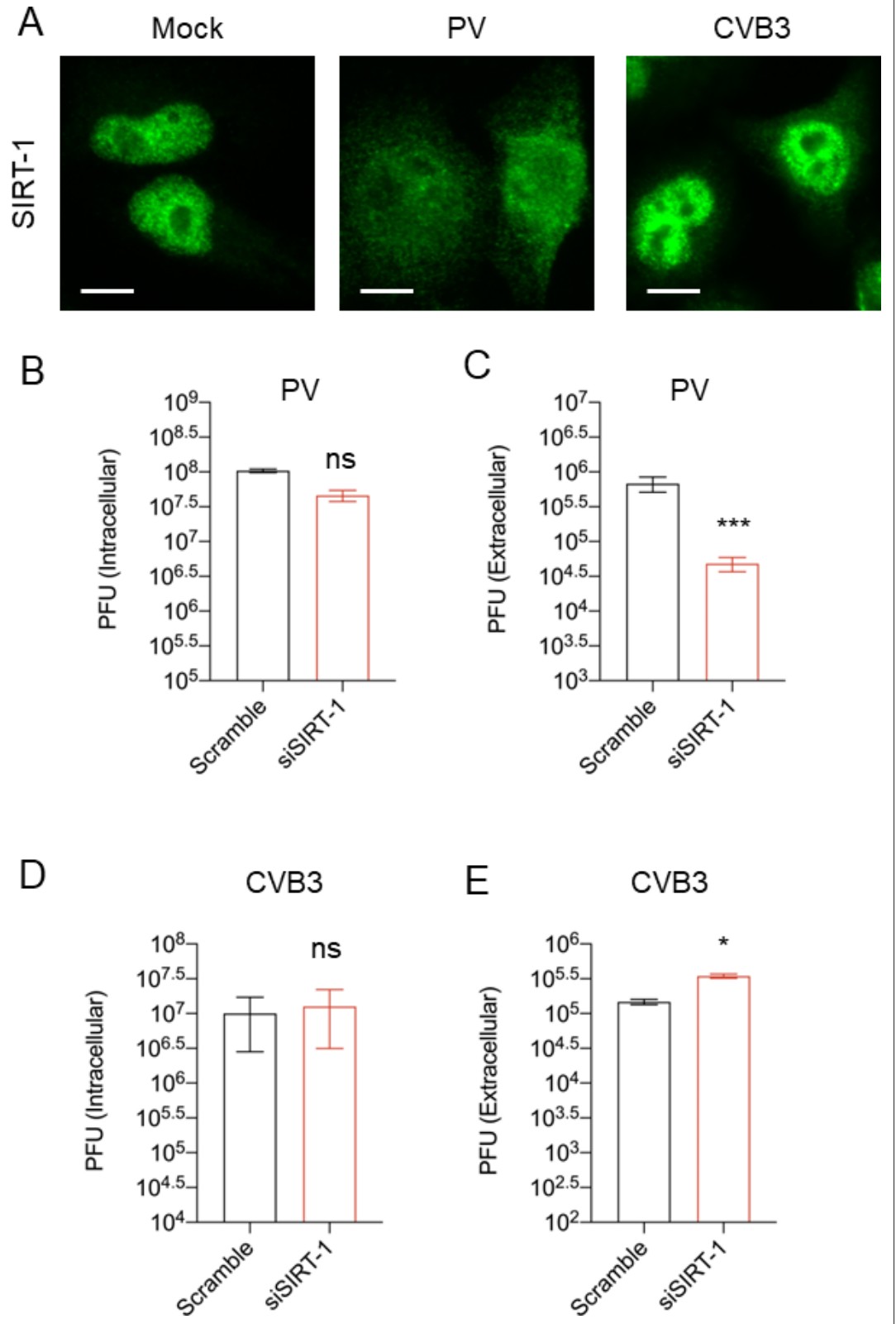

**Figure 5.** SIRT-1 is critical for poliovirus (PV), but not coxsackievirus B3 (CVB3), release from cells. (**A**) H1HeLa cells were mock-infected or infected with PV or CVB3 for 4 hr. The cells were fixed, and immunofluorescence analysis (IFA) was done against SIRT-1 MOI = 30. (**B, C**) Cells were transfected with scramble or SIRT-1 siRNAs for 48 hr before being infected with PV (MOI = 0.1) for 5 hr. Viral titers were determined by a plaque assay. (**D, E**) Cells were

*Figure 5 continued on next page*

*Figure 5 continued*

transfected and infected with CVB3 as in B. *n* = 3 independent experiments, and error bars represent mean ± standard error of the mean (SEM) (***p < 0.001; *p ≤ 0.05; ns = not significant). Scale bar = 7.5 μm.

The online version of this article includes the following figure supplement(s) for figure 5:

**Figure supplement 1.** EX527 pretreatment reduced poliovirus (PV), but not coxsackievirus B3 (CVB3) titers.

called non-lytic release. For instance, PV has been shown to escape intact cells through AWOL (autophagosome-mediated exit without lysis), in which virus-containing autophagosomes fuse with the plasma membrane to release virions (*Jackson et al., 2005*; *Richards and Jackson, 2012*). We show that knockout of ATG-7 reduces EV-D68 extracellular/EV titers, which is consistent with the AWOL model (*Figure 2C, D*). Hepatitis A virus (HAV), which is non-lytic, has been reported to exit cells in an enveloped form known as eHAV (*Feng et al., 2013*; *Rivera-Serrano et al., 2019*). Similar events, modulated through the autophagic pathway, have been shown for multiple enteroviruses (*Chen et al., 2015*; *Robinson et al., 2014*; *Sin et al., 2017*). We show here for the first time that EV-D68 can also exit intact cells non-lytically in EVs. This extracellular vesicle-mediated release of EV-D68 virions is, as we demonstrate here, dependent upon SIRT-1, a protein best known as a histone deacetylase and autophagy regulator (*Figure 7A*). However, our data suggest that SIRT-1's proviral activity is mediated through a role in repressing ER stress.

Studies have shown that SIRT-1 negatively regulates crucial ER stress-related proteins, including XBP1 (*Wang et al., 2011*). Therefore, knocking down SIRT-1 would be expected to permit ER stress induction, which, in turn, could attenuate EV-D68 non-lytic release. Consistent with this presumption, provoking ER stress through TG treatment, similar to SIRT-1 knockdown, markedly decreases EV-D68 extracellular titers in both H1HeLa and hSABCi cells (*Figures 3A and 4A*). Our drug-based epistasis analysis revealed that SIRT-1 and TG share common cellular targets since TG did not further reduce viral titers in SIRT-1 knockdown cells (*Figure 3B*).

Since TG binds SERCA2A and downregulates its activity leading to ER stress (*Lytton et al., 1991*), we investigated whether depleting SERCA2A will impact EV-D68 extracellular titers. As anticipated, knocking down SERCA2A reduces EV-D68 release. However, double knockdown of SIRT-1 and SERCA2A failed to appreciably decrease viral titers compared to SERCA2A or SIRT-1 knockdowns alone, indicating SERCA2A is a downstream target of SIRT-1. Intriguingly, we observed that the knockdown of SIRT-1 concomitantly reduced SERCA2A protein levels (*Figure 3E*), which is in agreement with a previous study (*Gorski et al., 2019*). Therefore, knocking down SIRT-1 decreases SERCA2A levels, allowing ER stress induction and decreasing EV-D68 release.

TG has been demonstrated to possess antiviral activity against many viruses using various mechanisms. For instance, TG attenuates Peste des petits ruminantsvirus (PPRV) and Newcastle disease virus (NDV) infection by restricting viral entry and viral protein synthesis (*Kumar et al., 2019*). In contrast, TG inhibits respiratory viruses such as Influenza and coronaviruses by provoking a robust interferon response (*Al-Beltagi et al., 2021*). Here, for the first time, we demonstrate that TG also inhibits EV-D68 infection (*Figures 3A and 4A*). TG's anti-EV-D68 activity seems unlikely to involve inhibition of viral entry, viral RNA replication, or inducing the type 1 interferon response since the drug treatment only marginally reduced EV-D68 intracellular titers (*Figures 3A and 4A*). Instead, TG's anti-EV-D68 activity involves limiting the non-lytic release of EV-D68 (*Figures 3A and 4A*).

Although exactly how TG attenuates EV-D68 non-lytic release is unclear, we hypothesize that TG reduces EV-D68 non-lytic release by inducing ER Stress. Consistent with this hypothesis, the knockdown of SERCA2A, which induces ER stress, also reduces EV-D68 release (*Figure 3D*). Since EV-D68 infection inhibits ER stress, as indicated by the decrease in XBP1 and BiP protein levels, and the lack of BiP induction when EV-D68-infected cells were treated with TG (*Figures 3F and 4B*), this suggests that ER stress is detrimental to EV-D68 release. Although we did not observe specific XBP1 or BiP cleavage products during EV-D68 infection, a recent study showed that PV and CVB3 induce cleavage of XBP1 at late time points during their infection (*Shishova et al., 2022*). However, the study did not determine the specific mechanism of XBP1 cleavage. Nonetheless, decreasing full-length XBP1 and BiP levels during infection may be a common strategy employed by enterovirus to avoid ER stress, which, as we have shown, is a negative for non-lytic release. While SIRT-1 is thought to negatively regulate ER stress by deacetylating XBP1, we found that the deacetylase inactive mutant (SIRT-1

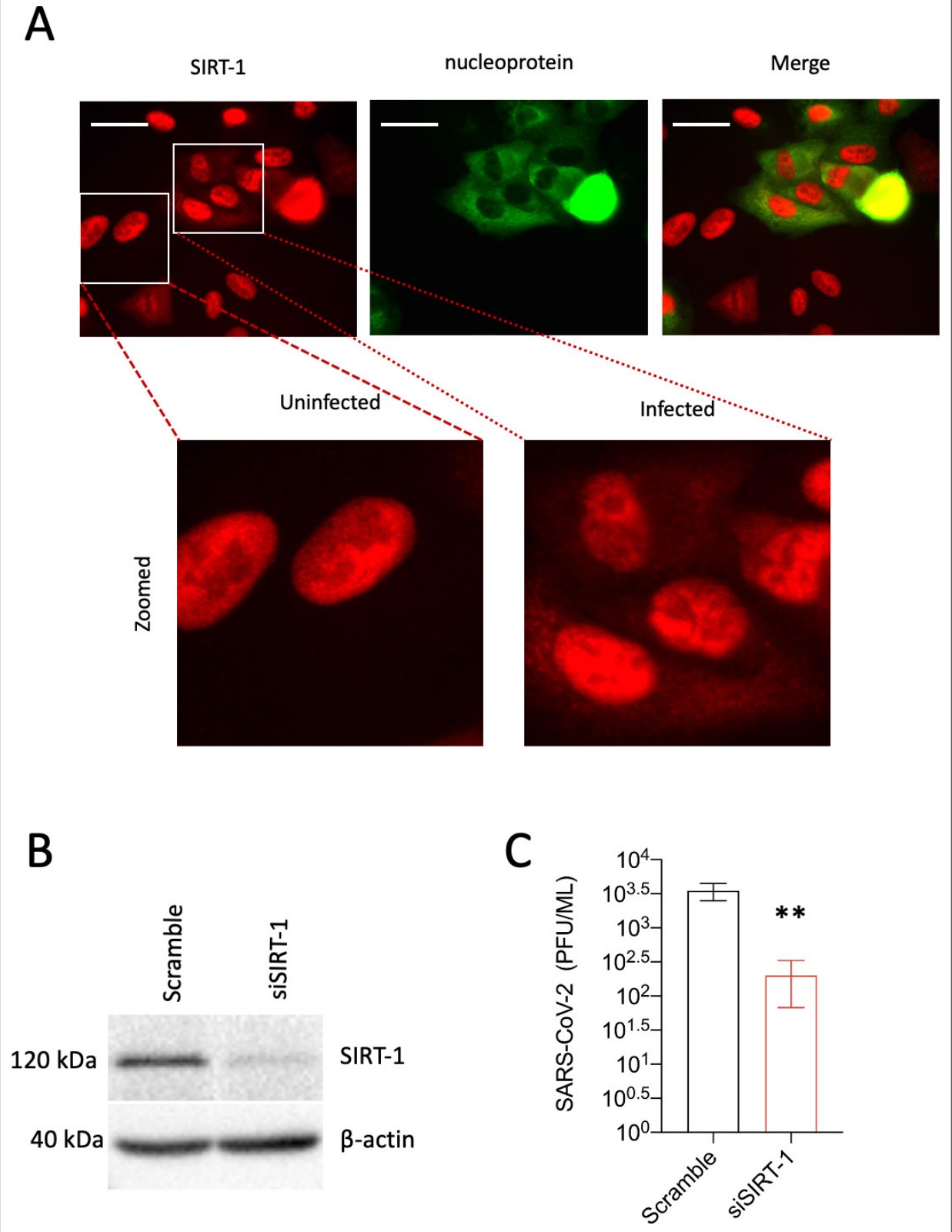

**Figure 6.** SARS-CoV-2 infection requires SIRT-1 and induces SIRT-1 translocation to the cytosol. (**A**) A549-ACE2 cells were infected with SAR-CoV-2 (MOI = 0.5) for 24 hr. The cells were fixed and subjected to immunofluorescence analysis (IFA) against SIRT-1 and SARS-CoV-2 nucleoprotein. (**B**) A549-ACE2 cells were transfected with SIRT-1 siRNA for 48 hr. Lysates were analyzed by western blot. (**C**) Cells were transfected as in B and infected (MOI 0.01) for 48 hr before being subjected to a plaque assay for viral titer measurement (\*\*p < 0.01). Scale bar = 30 μm. *n* = 3 independent experiments.

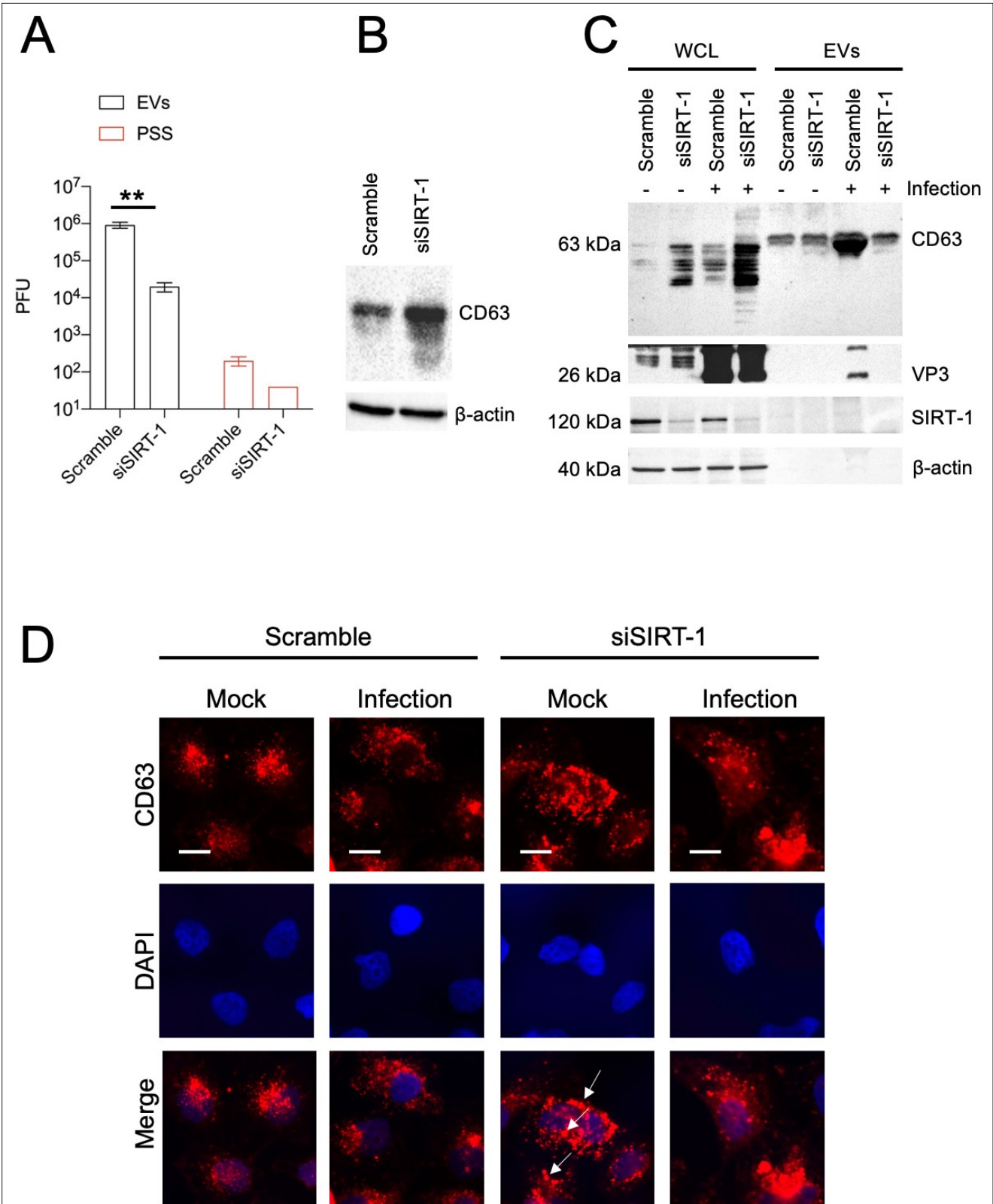

**Figure 7.** SIRT-1 KD reduces extracellular vesicle-mediated release of infectious enterovirus D68 (EV-D68) viral particles. H1HeLa cells were transfected with SIRT-1 and Scramble siRNAs for 48 hr. (**A**) Cells were infected with EV-D68 (MOI = 0.1) for 5 hr, and EVs were isolated for viral titer measurement by plaque assay. (**B**) Cells were transfected as in A, and the whole-cell lysates (WCL) were collected and prepared for western blot against CD63. (**C**) H1HeLa cells were transfected with the indicated siRNAs for 48 hr. The cells were either left uninfected or infected with MOI 30 of EV-D68 for 4 hr. The

*Figure 7 continued on next page*

*Figure 7 continued*

EVs and WCL were prepared for western blot against the indicated proteins. (**D**) Cells were plated on cover slides and transfected with scramble or SIRT-1 siRNA for 48 hr. The cells were then fixed and subjected to immunofluorescence analysis against CD63. Arrows indicate large CD63 puncta. Error bars indicate mean ± standard error of the mean (SEM) of three independent experiments. Unpaired Student's *t*-test was used for the statistical analysis (**p < 0.01). Scale bar = 6.5 μm.

The online version of this article includes the following figure supplement(s) for figure 7:

**Figure supplement 1.** SIRT-1 is not required for enterovirus D68 (EV-D68) entry and replication.

H363Y) increased EV-D68 release (*Figure 1C*). This suggests a non-deacetylase role of SIRT-1 in ER stress, at least during enterovirus infection.

SIRT-1 was previously demonstrated to be essential for MERS-CoV infection (*Weston et al., 2019*). Here, we demonstrate that the cellular protein is also crucial for SARS-CoV-2 infection. Our data show that, much like in the MERS-CoV study, the knockdown of SIRT-1 decreases SARS-CoV-2 release (*Figure 6C*), suggesting that SIRT-1 may be a shared host factor utilized by many Betacoronaviruses.

Our data show that a portion of nuclear SIRT-1 relocalizes to the cytosol upon infection by EV-D68 and PV (*Figures 1D, 4D, and 5A*). We also observe that SARS-CoV-2 infection causes SIRT-1 translocation to the cytosol (*Figure 6A*). These viruses require SIRT-1 for normal virus release. It will be interesting to examine whether other Betacoronaviruses alter the subcellular localization of SIRT-1 and whether SIRT-1 is important for their infection.

CVB3, which does not require SIRT-1, also fails to induce relocalization of the protein (*Figure 5A*). This, along with the finding that SIRT-1 is found on virus-induced extracellular vesicles, suggests to us a central role for SIRT-1 in constructing virus-containing vesicles for extracellular release – but only for some viruses. Why at least one enterovirus has evolved a SIRT-1-independent mechanism for release and what the mechanism might be will help understand how to target-specific viruses or broad classes of viruses to prevent their release from infected cells. While we believe SIRT-1 is a common regulator of virus release, it is not a universal one. Future work identifying a homolog, or paralog, protein playing a parallel role in CVB3 release, and understanding the relationship of CVB3 to ER stress, will undoubtedly shed light on the differences between these otherwise similar viruses.

In summary, our data show that SIRT-1 is essential for basal- and stress-induced autophagy and EV-mediated non-lytic release of EV-D68. We also demonstrated that SIRT-1 is crucial for releasing other important public health viruses, including SARS-CoV-2 and PV, indicating that SIRT-1 may be a common host factor regulating multiple viruses' release. Understanding how SIRT-1 regulates viral egress could open avenues for therapeutic intervention against many viruses.

## Materials and methods
### Cell culture, plasmids, and viruses

H1HeLa cells were purchased from ATCC (CRL-1958) and cultured in Dulbecco's Modified Eagle Medium (DMEM) supplemented with 10% fetal bovine saline, 1× penicillin/streptomycin, and 1× sodium pyruvate. The hSABCi-NS1.1 immortalized small airway cells were grown in PneumaCult-Ex Plus Basal Medium supplemented with PneumaCult-Ex Plus 50× supplement, 0.2× hydrocortisone, 1× penicillin/streptomycin, 1.25 μg/ml amphotericin B, and 0.5 mg/ml gentamycin. Cells were regularly tested for mycoplasma and authenticated by short tandem repeat DNA profiling (Genomics Core Lab, University of Maryland, Baltimore). The cells were incubated at 37°C in a 5% $CO_2$ incubator. The wild-type (Flag-SIRT1) and mutant (Flag-SIRT1 H363Y) SIRT-1 plasmids were purchased from Addgene and transfected into cells using Lipofectamine 2000. The transfection complex was replaced with basal media 6 hr post-transfection.

All work with SARS-CoV-2 was performed in a BSL3 laboratory and approved by our Institutional Biosafety Committee (IBC#00005484). Vero E6 cells overexpressing transmembrane serine protease 2 (TMPRSS2) (VeroT) (ATCC CRL-s1586) were cultured in DMEM medium (Quality Biological) supplemented with 10% (vol/vol) heat-inactivated fetal bovine serum (FBS) (Sigma), 1% (vol/vol) penicillin–streptomycin (Gemini Bio-Products) and 1% (vol/vol) L-glutamine (2 mM final concentration; Gibco). A549 cells overexpressing human angiotensin-converting enzyme 2 (hACE2, A549/hACE2) were generously provided by Dr. Brad Rosenberg (*Daniloski et al., 2021*). They were cultured in DMEM medium (Quality Biological) supplemented with 10% (vol/vol) heat-inactivated FBS (Sigma), and 1%

(vol/vol) penicillin–streptomycin (Gemini Bio-Products). For induction of autophagy by starvation before infection, cells were starved for 4 hr with the 'Axe' media (an amino acid deficient media that is widely used to induce autophagic flux) (*Axe et al., 2008*).

## Western blot

Cells were lysed using radioimmunoprecipitation assay buffer buffer supplemented with cOmplete Tablets Mini Protease Inhibitor Cocktail. The lysates were incubated on ice for at least 30 min before being clarified at 12,000 rpm for 30 min. The supernatants were transferred into Eppendorf tubes, and protein concentrations were determined by Bradford assay. Lysates were then boiled and loaded onto sodium dodecyl sulfate–polyacrylamide gel electrophoresis. Following transfer onto polyvinylidene difluoride (PVDF) membranes, the membranes were blocked in 5% skim milk for 1 hr, washed twice with Tris-buffered saline with 0.1% Tween 20 detergent (TBST), and stained with the following primary antibodies: anti-SQSTM1/p62, anti-LC, anti-β-actin at 1:1000 dilutions, anti-CD63 (1:250), anti-SERCA2A (1:500), and anti-XBP1 (1:500) overnight. The membranes were stained with the secondary antibodies for 1 hr at room temperature and imaged using the Chemidoc machine after two washes.

## Immunofluorescence analysis

The cells were fixed with 4% paraformaldehyde at room temperature for 20 min and then permeabilized with 0.3% Triton-X for 30 min. The cells were blocked with 3% bovine serum albumin for 1 hr on a shaker and incubated with primary antibodies at 1: 250 dilutions overnight at 4°C. The cells were washed twice with phosphate-buffered saline (PBS), incubated with the secondary antibodies (1:250), rewashed three times, and imaged with an ECHO Revolve fluorescence microscope.

## Extracellular vesicle isolation

Extracellular vesicles were isolated using the Invitrogen Total Exosome Isolation Reagent (from cultured cells). The reagent ties up water molecules and forces less soluble components, such as vesicles, out of the culture media, which can then be pelleted by a short centrifugation. In brief, 1 ml of cell culture supernatants were clarified at 2000 × *g* for 30 min. The supernatants were then transferred to Eppendorf tubes, and 500 µl of the exosome isolation buffer was added and incubated at 4°C overnight. At the end of the incubation, the tubes were centrifuged for 1 hr at 10,000 × *g*. The supernatants were transferred to new Eppendorf tubes, and the pellets were resuspended in PBS for western blot or plaque assay.

## RNA isolation and qPCR

TRIzol was used to isolate total RNA according to the manufacturer's instructions, and cDNA was synthesized using the Thermo Scientific RevertAid H Minus First Strand cDNA Synthesis Kit. KiCqStart SYBR qPCR Ready Mix was used to perform qPCR using the Fast Dx Real-Time PCR Instrument (Applied Biosystems). Primers specific to the 5' untranslated region (5'-TAACCCGTGTGTAGCTTGG-3' and 5'-ATTAGCCGCATTCAGGGGC-3') were used to amplify EV-D68, and gene expression was normalized to GAPDH and plotted as relative expression compared to the 0 hr infection-only time point.

## Plaque assay

The cells were washed twice with PBS for cell-associated titer determination and scraped in 1 ml PBS, after which they were subjected to three freeze–thaw cycles and added to H1HeLa cells for 30 min. Cells were then overlaid with a 1:1 ratio of 2× Menimum Essential Medium (MEM) and 2% agar for 48 hr before staining plaques with crystal violet. For extracellular titers, 1 ml of supernatants were collected and treated as the cell-associated titer without being subjected to freeze–thaw cycles.

## siRNA transfections

siRNAs were transfected into cells using lipofectamine as previously described. In brief, 200 nM of siRNA and 10 µl of Lipofectamine 2000 were separately incubated in Opti-MEM at room temperature for 5 min. The siRNAs and Lipofectamine were mixed and incubated for 20 min before being added to cells that were 40% confluent. The transfection complexes were replaced with growth media at 6 hr

post-transfection. The cells were then used for viral infection or western blot for transfection efficiency determination.

## Virus entry assays

Scramble and SIRT-1 knockdown cells were pre-chilled on ice for 30 min for viral binding assay. The cells were then infected with EV-D68 (MOI = 30) for 1 hr on ice. The inoculum was removed, and the cells were washed twice with PBS, scraped, freeze–thawed three times, and stored at −80°C for plaque assay.

For viral entry, scramble and SIRT-1 knockdown cells were similarly pre-chilled on ice for 30 min and then infected with EV-D68 for 30 min on ice. The unbound viral particles were washed off with PBS before shifting the cells to 37°C, allowing viral entry for 1 hr. The cells were finally washed with PBS, scraped into Eppendorf tubes, and prepared for plaque assay.

## SARS-CoV-2 titer determination by plaque assay

Plaque assays were performed as described previously (*Coleman and Frieman, 2015*). Briefly, 12-well plates were seeded with $2 \times 10^5$ VeroT cells/well one day before processing. On the day of processing, media was removed from the 12-well plates, and 200 µl of dilutions of virus stock or collected cell supernatants in DMEM were added to each well. Plates were incubated at 37°C (5% $CO_2$) for 1 hr with rocking every 15 min. Following incubation, 2 ml of plaque assay media, DMEM containing 0.1% agarose (UltraPure) and 2% FBS (Gibco), was added to each well and incubated for 48 hr at 37°C (5% $CO_2$). Following incubation, plates were fixed with 4% paraformaldehyde, stained with 0.25% crystal violet (wt/vol), plaques counted, and titers calculated as plaque-forming units (PFU)/ml.

## siRNA knockdown protocol for SARS-CoV-2 infection

siRNA knockdown assays were performed as described previously (*Weston et al., 2020*). Briefly, A549/hACE2 cells were seeded in 24-well cell culture plates one day before siRNA treatment. On the day of treatment, 4.4 µl Opti-MEM (Gibco) and 2.2 µl Oligofectamine (Thermo Scientific) were combined and incubated for 5 min at room temperature. This mixture was then added to 35.5 µl Opti-MEM and 0.8 µl of 50 µM siRNA and incubated for 20 min at room temperature. Following incubation, a further 177 µl of Opti-MEM was added to the transfection mixture, media were removed from cells, and 200 µl of transfection mixture was added. After a 4 hr incubation at 37°C/5% $CO_2$, 200 µl of DMEM (+20% FBS) was added to the cells resulting in a final concentration of 10% FBS. Cells were then incubated at 37°C/5% $CO_2$ overnight. Following incubation, cells were infected with SARS-CoV-2 (WA1, MOI = 0.01 for titer, MOI = 0.5 for IFA), and supernatants were collected 24 hr post-infection. SARS-CoV-2 titers from supernatants were determined by plaque assay.

## Statistical analysis

GraphPad Prism software (Version 7.03) was used for all statistical analyses, and values represent the mean ± standard error of the mean of at least three independent repeats. Student's *t*-test was used for comparison and a p-value of <0.05 was considered statistically significant.

## Materials availability statement

All materials used are available from the commercial and non-profit suppliers listed. Contact corresponding author for assistance.

## Acknowledgements

We thank Sohha Ariannejad for assistance and the members of the Jackson, Frieman, and Coughlan labs for thoughtful discussion. This work was funded by NIH/NIAID grants R01141359, R01104928 to WTJ and R21158134 to WTJ and MBF.

# Additional information

## Funding

| Funder | Grant reference number | Author |
|---|---|---|
| National Institute of Allergy and Infectious Diseases | R01141359 | Alagie Jassey<br>Katelyn Miller<br>William T Jackson |
| National Institute of Allergy and Infectious Diseases | R01104928 | Michael A Wagner<br>Ganna Galitska<br>William T Jackson |
| National Institute of Allergy and Infectious Diseases | R21158134 | James Logue<br>Stuart Weston<br>Matthew Frieman<br>William T Jackson |

The funders had no role in study design, data collection, and interpretation, or the decision to submit the work for publication.

## Author contributions

Alagie Jassey, Conceptualization, Data curation, Formal analysis, Validation, Investigation, Visualization, Methodology, Writing – original draft, Writing – review and editing; James Logue, Stuart Weston, Resources, Investigation; Michael A Wagner, Ganna Galitska, Resources, Investigation, Writing – review and editing; Katelyn Miller, Resources, Methodology, Writing – review and editing; Matthew Frieman, Resources, Supervision, Funding acquisition, Writing – review and editing; William T Jackson, Conceptualization, Resources, Data curation, Supervision, Funding acquisition, Investigation, Writing – original draft, Project administration, Writing – review and editing

## Author ORCIDs

Alagie Jassey ⓘ http://orcid.org/0009-0001-3669-3836
Michael A Wagner ⓘ https://orcid.org/0000-0002-4161-184X
Ganna Galitska ⓘ http://orcid.org/0000-0001-9365-6751
Katelyn Miller ⓘ http://orcid.org/0000-0002-8893-7011
William T Jackson ⓘ https://orcid.org/0000-0002-9832-0584

Joint Public Review: https://doi.org/10.7554/eLife.87993.3.sa1
Author Response https://doi.org/10.7554/eLife.87993.3.sa2

# Additional files

## Supplementary files
- MDAR checklist
- Source data 1. Western blot source data.

## Data availability

All data are available at https://osf.io/rgecf/.

The following dataset was generated:

| Author(s) | Year | Dataset title | Dataset URL | Database and Identifier |
|---|---|---|---|---|
| Jassey A, Logue J, Weston S, Wagner MA, Galitska G, Miller K, Frieman MB, Jackson WT | 2023 | SIRT-1 is required for release of enveloped enteroviruses | https://osf.io/rgecf/ | Open Science Framework, rgecf |

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
