## [Editor Report · eLife assessment]

The presence or absence of a surrounding envelope, previously a clear distinguishing feature of different viruses, has been blurred by the recent recognition that many so-called 'nonenveloped' viruses are released from cells as quasi-enveloped virions cloaked in host cell membranes. This mechanism of viral egress allows for non-lytic infection, and has potentially **important** implications for pathogenesis. In this manuscript, Jassey and colleagues provide **solid** evidence that the protein deacetylase SIRT-1 is required for the non-lytic release of enteroviruses in extracellular vesicles.

---

## [Referee Report · Joint Public Review]

The enteroviruses comprise a medically important genus in the large and diverse picornavirus family, and are known to be released without lysis from infected cells in large vesicles containing numerous RNA genome-containing capsids - a feature allowing for en bloc transmission of multiple viral genomes to newly infected cells that engulf these vesicles. SIRT-1 is an NAD-dependent protein deacetylase that has numerous and wide ranging effects on cellular physiology and homeostasis, and it is known to be engaged in cellular responses to stress and autophagy.

Jassey et al. show that RNAi depletion of SIRT-1 impairs the release of enterovirus D-68 (EV-D68) in EVs recovered from the supernatant fluids of infected cells using a commercial exosome isolation kit. The many functions attributed to SIRT-1 in the literature reflect its capacity to deacetylate various cell proteins engaged in transcription, DNA repair, and regulation of metabolism, apoptosis and autophagy. However, Jassey et al. make the surprising claim that the proviral role of SIRT-1 in promoting enterovirus release is not dependent on its deacetylase activity. Fig. S1C is crucial to this suggestion but it is less than completely convincing. It shows that both wild-type and mutant SIRT-1are massively over-expressed in the rescue experiment compared to the normal endogenous level of SIRT-1 expression. Moreover, the blots are heavily saturated, making it difficult to assess the relative expression of wild-type vs. mutant. In addition, Fig. S1B and Fig. 4C convincingly show that EX527, a small molecule inhibitor of the deacetylase activity of SIRT-1, inhibits extracellular release of the virus. This suggests that the deacetylase activity of SIRT-1 may in fact be required for the proviral effect of SIRT-1. This is a fundamentally important question that requires more investigation.

Fig. 6 shows how SIRT-1 knockdown impacts the release of enterovirus D68 in EVs recovered from cell culture supernatant using a commercial 'Total Exosome Isolation Kit'. The authors are appropriately cautious in describing the vesicles they presume to be isolated by the kit as simply 'extracellular vesicles', since there are multiple types of EVs with very different mechanisms of biogenesis, of which 'exosomes' are but one specific type. It would have been more elegant had the authors shown that SIRT-1 is required for EV-D68 release in detergent-sensitive vesicles with low buoyant density in isopycnic gradients, and to characterize the size and number of viral capsids in these vesicles by electron microscopy.

The authors claim that "reduction of SIRT-1 attenuates the release of virus-loaded CD63-positive EVs" but they never actually show that the vesicles containing EV-D68 are in fact CD63-positive. Can a CD63 pulldown immunoprecipitate EV-D68 capsid proteins or viral RNA? This is important since CD63 is strongly associated with exosomes released from cells through the multi-vesicular body pathway, which are distinct from the LC3-positive EVs released by secretory autophagy that have previously been associated with enteroviruses.

The authors claim "that most EV-D68 is released non-lytically in an enveloped form" but they show data from only from early time points following infection (5 or 6 hrs post-infection) - prior to cell lysis. It would have been interesting to see a more complete temporal analysis, and to know the overall proportion of virus released in EVs versus lytic release of nonenveloped virus.

Fig. 1D indicates that a small fraction of SIRT-1 leaks from the nucleus in EV-D68 infected cells. The authors suggest this is due to targeted nuclear export, rather than simply leaky nuclear pores which are well known to exist in enterovirus-infected cells, but the evidence for this is questionable. The authors present similar fluorescent microscopy data showing inhibition of TFEB export in leptomycin-B treated cells in Fig. S2A in support of their claim that there is specific SIRT-1 export, but there is equivalent residual TFEB and SIRT-1 in the cytoplasm of the treated cells. Quantitative immunoblots of cytoplasmic and nuclear cell fractions might prove more compelling.

---

## [Author Response]

The following is the authors’ response to the original reviews.

**Reviewer #1 (Public Review):**
The enteroviruses comprise a medically important genus in the large and diverse picornavirus family, and are known to be released without lysis from infected cells in large vesicles containing numerous RNA genome-containing capsids - a feature allowing for en bloc transmission of multiple viral genomes to newly infected cells that engulf these vesicles. SIRT-1 is an NAD-dependent protein deacetylase that has numerous and wide ranging effects on cellular physiology and homeostasis, and it is known to be engaged in cellular responses to stress and autophagy.Jassey et al. show that RNAi depletion of SIRT-1 impairs the release of enterovirus D-68 (EVD68) in EVs recovered from the supernatant fluids of infected cells using a commercial exosome isolation kit. The many functions attributed to SIRT-1 in the literature reflect its capacity to deacetylate various cell proteins engaged in transcription, DNA repair, and regulation of metabolism, apoptosis and autophagy. However, Jassey et al. make the surprising claim that the proviral role of SIRT-1 in promoting enterovirus release is not dependent on its deacetylase activity. Fig. S1C is crucial to this suggestion, as it is said to show that reconstituting expression with a catalytically-inactive mutant can rescue virus release from SIRT-1 depleted cells. However, no information is provided concerning the levels of endogenous and ectopicallyexpressed SIRT-1 proteins in this experiment, making it very difficult to interpret the results. Is the mutant SIRT-1 protein expressed at a higher level than the non-mutant protein? Is there a 'sponging' effect with these transfections that lessens the siRNA efficiency and reduces knockdown of the endogenous protein? Fig. S1B and Fig. 4C convincingly show that EX527, a small molecule inhibitor of the deacetylase activity of SIRT-1, inhibits extracellular release of the virus. This suggests that the deacetylase activity of SIRT-1 is in fact required for the proviral effect of SIRT-1. This is a fundamentally important question that will require more investigation.

We have included western blot data (Fig. S1D), which shows comparable levels of expression between the wild-type and mutant SIRT-1 constructs as well as the endogenous SIRT-1. While both constructs partially rescued EV-D68 titers in SIRT-1 knockdown cells, only the wild-type construct rescued SERCA2A protein levels, indicating that SIRT-1 deacetylase activity is required for SERCA2A expression but not for EV-D68 infection.

Fig. 6 shows how SIRT-I knockdown impacts the release of enterovirus D68 in EVs recovered from cell culture supernatant using a commercial 'Total Exosome Isolation Kit'. The authors should describe the principle this kit exploits to isolate 'exosomes' (affinity isolation?) and specify which antibodies it involves (anti-phosphatidylserine, anti-CD63, others?) This could impact the outcome of these experiments, and moreover is important to include in the longterm scientific record. The authors are appropriately cautious in describing the vesicles they presume to be isolated by the kit as simply 'extracellular vesicles', since there are multiple types of EVs with very different mechanisms of biogenesis, of which 'exosomes' are but one specific type. It would have been more elegant had the authors shown that SIRT-1 is required for EVD68 release in detergent-sensitive vesicles with low buoyant density in isopycnic gradients, and to characterize the size and number of viral capsids in these vesicles by electron microscopy.

We have added a description of the Total Exosome Isolation Kit principle to the materials and methods. The reagent, in brief, ties up water molecules and forces less soluble components, such as vesicles, out of the culture media, which can then be pelleted by centrifugation. The purity and size distribution of exosomes isolated with this kit is comparable to ultracentrifugation.

Fig. 6 shows that SIRT-1 depletion upregulates CD63 expression, but has no apparent impact on the release of CD63-positive 'EVs' from uninfected cells. EV-D68 infection also upregulates CD63 expression in SIRT-1 replete cells, and in this case, increases the release of CD63-positive EVs. The combination of infection and SIRT-1 depletion massively upregulates CD63 expression, but appears to eliminate the enhanced release of CD63-positive EVs resulting from infection alone. These are interesting results, from which the authors infer CD63 is associated with EVs containing EV-D68. But, do we know this? Can a CD63 pulldown immunoprecipitate EV-D68 capsid proteins or viral RNA? CD63 is strongly associated with exosomes released from cells through the multi-vesicular body pathway, which are distinct from the LC3-positive EVs released by secretory autophagy that have previously been associated with enteroviruses. The authors suggest that 'knockdown of SIRT-1 may prevent the exocytosis of CD63-positive EVs", but this is a very broad claim (and not really demonstrated by Fig. 6): it requires a clearer definition of what the authors mean by 'exocytosis' and a much more detailed analysis of the size and buoyant density of EVs released in a SIRT-1-dependent process.

We have toned down this suggestion, which sets up our logic for what is now Figure 7 but we agree does not prove the specific nature of these vesicles.

The authors suggest that almost all EV-D68 released from infected cells is released without cell lysis in EVs. However, they generally show data from only a single time point following infection (5 or 6 hrs post-infection). It would have been interesting to see a more complete temporal analysis, and to know whether a high proportion of virus continues to be released in EVs, or if it is swamped out ultimately by lytic release of nonenveloped virus.

In these cells, very little virus is released at earlier timepoints, and after 6hpi it is difficult to analyze virus release because of cell detachment and lysis. In a future publication we will use less susceptible cells to analyze a time course of release.

Fig. 1D indicates that a small fraction of SIRT-1 leaks from the nucleus in EV-D68 infected cells. The authors suggest this is due to targeted nuclear export, rather than simply leaky nuclear pores which are well known to exist in enterovirus-infected cells. The authors present similar fluorescent microscopy data showing inhibition of TFEB export in leptomycin-B treated cells in Fig. S2A in support of their claim that this is specific SIRT-1 export, but these data are far from convincing - there is equivalent residual TFEB and SIRT-1 in the cytoplasm of the treated cells.Quantitative immunoblots of cytoplasmic and nuclear cell fractions might prove more compelling.

We have changed the text to remove the word “block” and instead suggest that there is inhibition, given the difference we observe with and without leptomycin-B.

Finally, the authors should be more specific in describing the viruses they have studied (EV-D68 and PV). It would be preferable to describe these as 'enteroviruses' (including in the title of the manuscript), rather than more broadly as 'picornaviruses'. There is no certainty that the requirement for SIRT-1 in non-lytic release of virus extends to hepatoviruses or other picornaviral genera, for which mechanisms of nonlytic release may be quite different.

We have made this change and thank the reviewer for pointing this out.

**Reviewer #2 (Public Review):**
The authors aimed to connect SIRT-1 to EV-D68 virus release through mediating ER stress. They are successful in robustly connecting these pathways experimentally and show a new role for SIRT-1 in EV-D68 infection. These results extend to additional viruses, suggesting role(s) for SIRT-1 in diverse virus infection.The authors note that EV-D68 does not significantly impact SIRT-1 protein levels (Fig 1E and F), though this has been described for other picornaviruses (Xander et al., J Immunol 2019; Han et al., J Cell Sci 2016; Kanda et al Biochem Biophys Res Commun 2015). This may be of interest to note in the manuscript.

We have cited the above papers in the manuscript and thank the reviewer for these suggestions.

The data regarding CVB3 (Fig S4) are especially interesting because they show no discernable impact on infection. The manuscript should describe this further and perhaps speculate on potential reasons. Could it be due to inefficient knockdown?

We have shown that both genetic and pharmacological inhibition of SIRT-1 does not significantly alter CVB3 titers. We do not think this is due to inefficient knockdown since the CVB3 and PV experiments were done concurrently. We are currently investigating why CVB3 responds differently from EV-D68 and PV.

SIRT-1 (and other sirtuins) have been linked to an innate interferon response. Are any of the phenotypes observed here due to IFN responses? The use of H1HeLa cells would suggest this is not the case.

We think this is unlikely because H1HeLas are not IFN-competent and the knockdown of SIRT1 did not significantly alter viral RNA replication

**Reviewer #1 (Recommendations For The Authors):**
In Fig. 1, it would be informative to show an immunoblot of the protein in knockdown vs control cells (this is shown in different experiments in Fig. 2A and 3C, with variable degrees of knockdown efficiency, but ideally should be shown here also).

The knockdown efficiency of SIRT-1 is now shown in Fig. S1D. We thank the reviewer for this suggestion.

Why is the extracellular virus titer in the control cells in Fig. 1C so much lower (over a 1.5 logs) than in Fig. 1B? Has the plasmid transfection induced an innate immune response, and could this be confounding the experiment?

We think this is due to stress induced by transfection and not an innate immune response, since H1Hela are not interferon competent.

SIRT-1 is recognized to have a regulatory role in autophagy, but the author's claim that it is"essential for stress induced and basal autophagy" would be strengthened by including in Fig.2B control images of starved and CCCP-treated cells.

LC3 lipidation and p62 degradation are the hallmarks of autophagy initiation and flux, which are shown in Fig. 2A. The goal of Fig. 2B was to verify the impact of SIRT-1 knockdown in restricting basal autophagic degradation. We will examine the effect of starvation and CCCP treatment in future studies. We thank the reviewer for understanding.

The BiP immunoblot shown in Fig. 4B does not support the claim that 'TG [thapsigargin] treatment induced BiP protein levels' whereas 'EV-D68 infection reduced BiP levels...suggesting that EV-D68 blocks ER stress.' The apparent differences in BiP expression are minimal and of questionable biological significance.

We have consistently observed a reduction in BiP levels during EV-D68 infection in both hSABCi-NS1.1 as indicated in Fig. 4B and H1HeLa (see Author response image 1), which is consistent with an ER stress blockade during EV-D68 infection.

**Author response image 1. sa2fig1:** 

Minor comments:1. The variable and wide-ranging scale of the y-axis in Figs. 1A-C and S1 is distracting, exaggerates small differences, and makes it difficult to assess the magnitude of differences in virus titers. The scale should be standardized and held constant in graphs showing results from similar types of experiments.

Our graphs are plotted based on the viral titers from experiments, mostly done on different days. We are confident that the variabilities in the y-axis do not affect the statistical analyses.

1. The number and types of (technical or biological?) of experimental replicates should be indicated in the figure legends. Ideally, each replicate should be individually plotted in graphs.

All experiments are repeated at least three times unless otherwise indicated. We have added this information to the figure legends.

1. Fig. S5C - how many replicates were done, and is there a statistically significant difference in viral RNA abundance at the last time point?

The experiment was done three times, twice with a low MOI (0.1) and once with a high MOI (30). There is no statistical difference at the last time point as shown in the graphs in Author response image 2.

**Reviewer #2 (Recommendations For The Authors):**
Figure 1D would benefit from staining for viral replication compartments (J2, for instance) to correlate the amount of viral dsRNA with nuclear egress of SIRT-1. Similar data would benefit Figure 5A. The data in Figure S5 suggests that most, but not all cells, are infected, so having this control seems important for their IFA experiments.

SIRT-1 dsRNA staining for EV-D68 infection is shown in Fig. S5A and all cells appear to be infected. The IFA data (Author response image 3) shows dsRNA staining of CVB3-infected cells.

**Author response image 3. sa2fig3:** 

Are EVs not released as efficiently with SIRT-1 knockdown? The authors show that knockdown reduces CD63 levels in purified EVs, but this could be explained if exosomes are not generated as robustly with SIRT-1 knockdown.

We don’t want to use the word “exosomes” since their definition is very specific, and only use it once in our manuscript, to describe known membrane associations of CD63. We do not think SIRT-1 knockdown affects the intracellular generation of EVs, since depleting SIRT-1 leads to the buildup of CD63 positive signals in the whole cell lysates compared to the scramble control (Fig. 7B and C). Instead, our data suggest that SIRT-1 regulates the release of EVs during EV-D68 infection.

Labels of graphs for "Infection" versus treatment ("TG" or "EX527") is unclear. All samples are presumably infected, so perhaps the authors meant to label these diagrams as untreated.

We have made the changes in the labels and thank the reviewer for helping make these graphs more clear.

The induction of ER stress with TG and repression of stress with EV-D68 infection is clear from BiP western blots. Are BiP levels reduced in SIRT-1 knockdown cells? Their data with TG treatment and knockdown suggests this may be possible.

We have not examined the impact of SIRT-1 knockdown on BiP protein levels. But since SIRT1 KD increases ER stress, as evidenced by a reduction in SERCA2A levels (Fig. 3C and E), we would expect an increase in BiP levels in SIRT-1 depleted cells.

Would the authors expect TG to reduce EVs with EV-D68 as well? Presumably, combination of TG with SIRT-1 would reduce EVs similar to the results shown in Figure 6C. They mention in the discussion that TG and SIRT-1 "share common cellular targets" so it would be interesting to determine if TG acts similar to SIRT-1 knockdown with regard to EVs.

We think TG will similarly reduce EVs in EV-D68-infected cells, and we are currently testing this hypothesis.

Because of the inclusion of the SARS-CoV-2 data and mention in the abstract, it may be appropriate to include that data (Fig S7) in the main figures. The authors mention SIRT-1 as important to MERS-CoV infection in the introduction, but SIRT-1 has been implicated in RNA virus infection, including picornaviruses (noted above). The expansion of this section to provide additional context would benefit the introduction and discussion.

We have moved the former Fig. S7 to the main manuscript as Fig. 6.